# MLLM-For3D: Adapting Multimodal Large Language Model for 3D Reasoning Segmentation

Jiaxin Huang[1]    Runnan Chen[2,†]    Ziwen Li[1]    Zhengqing Gao[1]    Xiao He[4]    Yandong Guo[4]
Mingming Gong[1,3]    Tongliang Liu[1,2,†]
[1]MBZUAI    [2]The University of Sydney    [3]The University of Melbourne    [4]AI2Robotic

## Abstract

Reasoning segmentation aims to segment target objects in complex scenes based on human intent and spatial reasoning. While recent multimodal large language models (MLLMs) have demonstrated impressive 2D image reasoning segmentation, adapting these capabilities to 3D scenes remains underexplored. In this paper, we introduce **MLLM-For3D**, a simple yet effective framework that transfers knowledge from 2D MLLMs to 3D scene understanding. Specifically, we utilize MLLMs to generate multi-view pseudo-segmentation masks and corresponding text embeddings, then unproject 2D masks into 3D space and align them with the text embeddings. The primary challenge lies in the absence of 3D context and spatial consistency across multiple views, causing the model to hallucinate objects that do not exist and fail to target objects consistently. Training the 3D model with such irrelevant objects leads to performance degradation. To address this, we first filter irrelevant views using token attention. With these reliable pseudo-labels, we develop a token-for-Query approach for multimodal semantic alignment, enabling consistent identification of the same object across different views. Moreover, we introduce a spatial consistency strategy to enforce that segmentation masks remain coherent in the 3D space, effectively capturing the geometry of the scene. Extensive evaluations of various challenging indoor scene benchmarks demonstrate that, even without labeled 3D training data, **MLLM-For3D** outperforms existing 3D reasoning segmentation methods, effectively interpreting user intent, understanding 3D scenes, and reasoning about spatial relationships.

## 1 Introduction

Understanding user intent and reasoning about the 3D spatial context are crucial for real-world vision applications [38, 39, 31, 43, 58], including embodied AI, autonomous driving, and augmented/virtual reality. Recent advances in 3D scene understanding, particularly in multimodal learning [20, 69, 22, 7, 21, 46, 9, 16], have spurred the development of 3D point cloud-based Large Language Models (3D-LLMs). These methods allow systems to infer implicit goals, localize objects in complex environments, and interact seamlessly with users. Compared to conventional segmentation, reasoning segmentation poses a more complex challenge, requiring deeper levels of semantic understanding and the ability to handle underspecified or context-dependent queries.

The concept of reasoning segmentation was first introduced in the 2D domain by LISA [36], which employed the mask-as-embedding paradigm to fine-tune large language models using abundant $\langle 2D, 3D \rangle$ paired data. Based on its success in 2D domains, initial attempts have been made to adapt this paradigm to 3D tasks [19, 23, 30]. However, these approaches have the prohibitive cost of generating high-quality pairs $\langle 3D, \text{text} \rangle$, which often involve labor intensive manual annotation or

---

[†]**Corresponding authors**
[‡]**Code available at:** `https://github.com/tmllab/2025_NeurIPS_MLLM-For3D`

39th Conference on Neural Information Processing Systems (NeurIPS 2025).

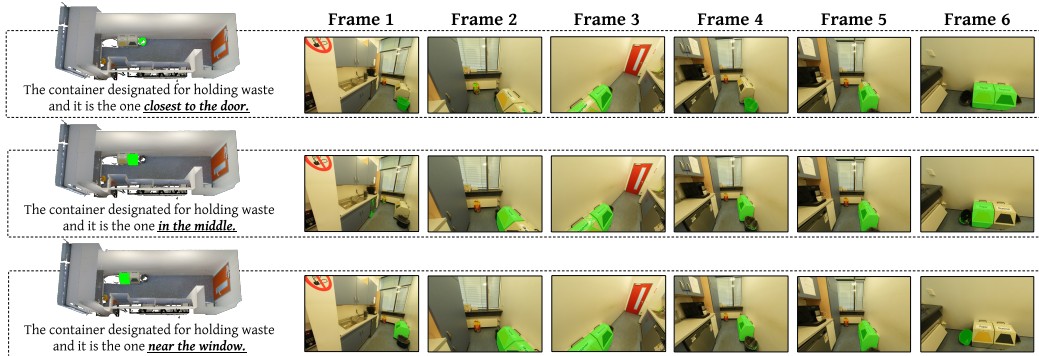

Figure 1: For the same scene, we present three different queries and display the 2D reasoning segmentation results on the same set of frames, illustrating how the model responds to varying instructions.

computationally expensive synthesizing (e.g., by GPT-4) [1]. A natural question thus arises: *Given a 3D point cloud accompanied by multiple posed RGB views, can we exploit pre-trained 2D reasoning segmentation models to approximate 3D labels?*

Our pilot study illustrates both the promise and the limitations of the idea (Figure 1). We evaluate the performance of 2D reasoning segmentation models, LISA [36] on the ScanNet++ dataset [64], by providing the model with different views corresponding to a 3D scene along with an implicit language instruction. Each frame is processed independently, generating both a reasoning response and a 2D binary segmentation mask. Ideally, the model should consistently localize the target object across all frames and infer its spatial relationships (e.g., nearest door) within the scene. However, as shown in Figure 1, there are two key limitations to the reasoning of 2D models in 3D: (i) *False positives in invisible views*: A 2D MLLM processing a single view might hallucinate or mistakenly segment objects that are described by the instruction but not visible in that particular view. Without 3D awareness, the model cannot distinguish between visible and occluded target objects, leading to incorrect mask predictions on some views. (ii) *Multi-View prediction inconsistency* occurs when the model lacks a mechanism to ensure spatial alignment of predictions across views, thereby degrading the performance when we aggregate multi-view predictions into 3D.

In this paper, we introduce **MLLM-For3D**, a simple yet effective framework that transfers 2D MLLM reasoning capabilities to 3D scene understanding. Specifically, we use a pre-trained MLLM to generate multi-view pseudo-segmentation masks and corresponding text embeddings. These masks are then unprojected into 3D space and aligned with the textual information to supervise the learning of the 3D model, eliminating the need for explicit 3D annotations. To address the critical issue of cross-view inconsistencies, we incorporate a **spatial consistency strategy** into the mask generation process, ensuring that the latent space remains coherent and mitigating object hallucinations. Specifically, we enforce latent space consistency by aggregating the per-view predictions via an attention-based fusion module. In this module, for a given 3D point visible in multiple views, the contribution of each view is weighted by its reliability and semantic similarity to a unified query derived from the [SEG] token embeddings. Furthermore, we propose a **token-for-Query** mechanism that consistently binds the same object identity across different views, enhancing the ability of the 3D model to interpret implicit user instructions and reason about spatial relationships.

MLLM-For3D is evaluated on three challenging benchmarks and shows that it achieves state-of-the-art performance in 3D reasoning segmentation tasks even without the need for any 3D annotations, which achieves about 55% higher mIoU than the previous methods. By effectively transferring 2D MLLM reasoning capabilities to 3D, our framework exhibits strong robustness to ambiguous queries, improved spatial reasoning, and superior segmentation performance.

The key contributions of our work are summarized as follows.

- We propose a simple, yet effective framework to adapt 2D MLLMs for 3D reasoning segmentation, eliminating the need for manual 3D annotations.

- We introduce a novel alignment mechanism that binds token embeddings to specific queries, ensuring consistent segmentation of the same object across views.

- We integrate a spatial consistency strategy to refine multi-view pseudo-segmentation masks, reducing the presence of hallucinated objects across frames.

- Extensive evaluations of two challenging indoor scene benchmarks demonstrate that MLLM-For3D outperforms existing 3D reasoning segmentation methods, even without any 3D labeled training data.

## 2 Related Works

### 2.1 Reasoning Segmentation

Reasoning segmentation is first introduced by LISA [36], which integrates a multimodal LLM (e.g., LLaVA [40]) with the Segmentation Anything Model (SAM) [35] to handle complex and *implicit* instructions in 2D images. PixelLM [49] builds on this paradigm by adopting a lightweight decoder and segmentation codebook for multi-object reasoning segmentation. LLM-Seg [54] uses SAM to propose candidate masks and allows the LLM to reason which mask fits the query. VISA [61] and VideoLISA [2] extend these approaches to video data, addressing temporal coherence and object tracking. FAST [51], an agent-based pipeline, further refines segmentation masks by iteratively identifying and masking key objects. These advancements in 2D [60, 67, 56, 53, 26] demonstrate the value of combining segmentation with LLM reasoning: models can interpret rich instructions and produce the corresponding mask, which is not possible with traditional segmentation alone.

**In 3D Domains**, PARIS3D [33] and Reasoning3D [8] focus on part segmentation with explanatory capabilities for individual objects, leaving scene-level reasoning tasks relatively unexplored. More recently, SegPoint [19], Reason3D [23], and MORE3D [30] have adapted the embedding-as-mask paradigm from LISA, aiming to unify multiple 3D tasks through human-like instructions. In parallel, Point-Bind and Point-LLM [16] extend 3D understanding to the multi-modal domain by aligning point clouds with images, language, audio, and video, and further enabling 3D large language models to follow multi-modal instructions. Despite these advances, such methods typically rely on large-scale $\langle 3D, text \rangle$ training data or parameter-efficient fine-tuning of LLMs, both of which are computationally costly. In contrast, our approach alleviates this problem by distilling reasoning capabilities and semantic knowledge from 2D MLLMs into a 3D model, enabling label-free 3D reasoning segmentation without any 3D supervision.

### 2.2 Label-Free 3D Scene Understanding

**Open-Vocabulary and Zero-Shot Approaches.** To alleviate the annotation burden, several label-free scene understanding methods [45, 44, 29, 25] leverage *vision foundation models* for zero-shot 3D segmentation. OpenScene [45] employs 2D open vocabulary segmentors [37, 15] to align pixel-level embeddings with 3D points, allowing object category recognition for unseen classes. CLIP2Scene [6] employs MaskCLIP [68] to obtain pixel-aligned features for annotation-free and label-efficient scene understanding. ConceptFusion [28] and CLIP-FO3D [65] further explore the acquisition of pixel-aligned knowledge through the extraction of dense region-level features using CLIP [47] and multi-view feature fusion. PLA [13] proposed a language-driven 3D scene understanding paradigm, which obtains point-language paired data through image captioning by visual-language foundation models for training 3D backbones. Similarly, RegionPLC [63] and Lowis3D [14] build point-caption pairs by projecting 2D visual-language features onto 3D geometry. Methods like OVIR3D [42] and MaskClustering [62] merge zero-shot 2D masks with 3D semantics for instance segmentation. Recent efforts [5, 29] also combine different foundation models (e.g., LLaVa-1.5 [40] and SEEM [70]) to unify zero-shot 2D embeddings and 3D point features, demonstrating strong category expansion in 3D. In parallel, LERF [34] introduces a 3D language grounding method that distills CLIP embeddings into NeRF volumes by optimizing a multi-scale language field through volume rendering and enforcing multi-view consistency.

**Label-Free 3D Reasoning Segmentation.** While these label-free strategies effectively handle open-vocabulary classes, their prompts remain relatively straightforward, limiting their ability to interpret more nuanced or context-heavy queries. In this work, we target *implicit* user prompts that require both semantic and spatial comprehension. Our framework, **MLLM-For3D**, inherits the reasoning

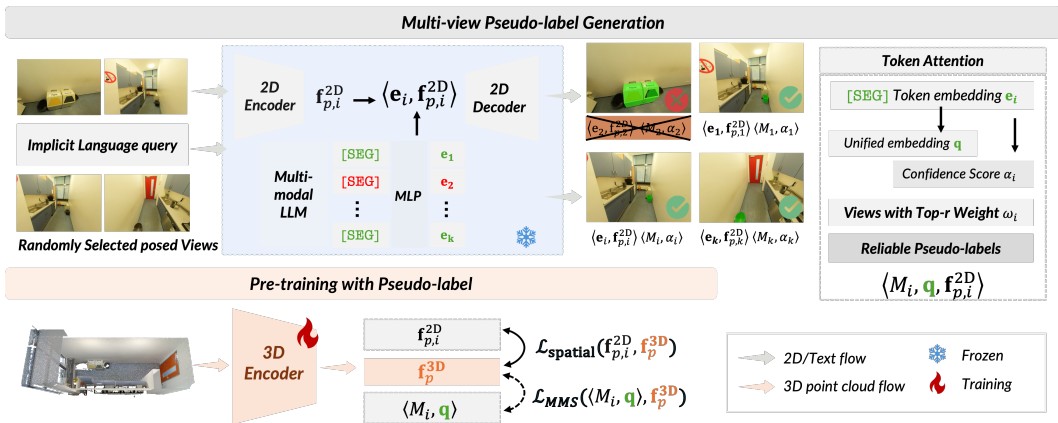

Figure 2: Overview of the proposed **MLLM-For3D** framework. We adapt multimodal large language models (MLLMs) for 3D reasoning segmentation by generating multi-view pseudo-labels and filtering irrelevant views via token attention. During the training phase, we enforce cross-view consistency via a spatial consistency strategy and align an unified embeddings $\mathbf{q}$ with 3D per-point feature $\mathbf{f}_p^{3D}$ via a multimodal semantic loss, enabling consistent object identity binding across views.

capabilities of 2D MLLM and applies them to 3D *without any annotations*, making it both *scalable* and *intuitive* in real-world scenarios.

## 3 Methodology

We propose MLLM-For3D, a label-free framework that adapts a 2D MLLM for 3D reasoning-based segmentation. As illustrated in Figure 2, a frozen MLLM and a 2D segmentation model are jointly used to generate multi-view pseudo-labels (2D binary masks $M_i$ and associated [SEG] token embedding $\mathbf{e}_i$) from randomly selected posed views. However, not all views capture the queried object (discussed in Section 1). To address this, we introduce an attention-based view filtering mechanism to select reliable views. Finally, a 3D segmentation network is trained under pseudo-label supervision, incorporating both semantic alignment and spatial consistency across views. All 2D models remain frozen, and only the 3D network parameters are optimized.

### 3.1 Multi-View Pseudo-Label Generation

**MLLM+SAM for Per-View Segmentation.** For each 3D scene, we assume a set of posed RGB images covering the scene, along with a textual query implicitly describing the target object. Previous multi-view 3D understanding approaches [57, 17, 18] demonstrate that view selection and aggregation are crucial for robust 3D perception. Therefore, we randomly select $k$ camera views and feed each image-text pair into a frozen MLLM to produce a special [SEG] token embedding $\mathbf{e}_i$ (for the $i$-th view), which semantically represents the queried segment in that view. The [SEG] embedding is then passed to the integrated SAM decoder within LISA to generate a binary mask $M_i$ and a confidence score $\alpha_i$ (predicted IoU). Repeating this process yields a collection of candidate 2D masks $M_i$, each hypothesizing the object's location based on 2D MLLM reasoning.

**View Filtering via Token Attention.** Since not all selected views contain the object, some $M_i$ may be empty or incorrect. To mitigate this, we first discard masks with very low confidence or area and then compute an attention weight $\omega_i$ to quantify each view's reliability. Each remaining embedding $\mathbf{e}_i$ contributes to a unified query embedding $\mathbf{q}$ via attention-weighted fusion. The attention weight $\omega_i$ reflects both mask confidence and semantic alignment: Formally, we define an attention score using the similarity of the dot product between $\mathbf{e}_i$ and $\mathbf{q}$ in a shared latent space. Let $s_i = \mathbf{e}_i \cdot \mathbf{q}$ denote the semantic alignment of the prediction of view $i$ with the unified embedding. We then set $\omega_i \propto \alpha_i \cdot \max(0, s_i)$ and normalize $\omega_i$ in all views so that $\sum_i \omega_i = 1$. Views with higher semantic consistency and clearer object visibility receive larger weights, while noisy or occluded views are effectively suppressed. This token attention mechanism thus reweights each view's contribution during 3D fusion, ensuring that only semantically coherent masks dominate the final pseudo-label set.

During fusion, each unprojected 3D mask is scaled by its corresponding token attention weight. The final prediction is the weighted average of these masks, where weights reflect the semantic confidence of each view.

While additional input views can provide more context, we observe that naïvely increasing view count beyond 4 degrades performance due to occlusions, inconsistent reasoning, and hallucinated [SEG] tokens from partially visible objects. Our token attention mechanism mitigates this by weighting each view based on semantic alignment, effectively filtering informative views.

## 3.2 Training with Pseudo-Labels

With reliable multi-view pseudo-labels, we train a 3D segmentation model that learns to localize the target object in the point cloud while enforcing cross-view semantic alignment and spatial consistency. The model encodes per-point feature $\mathbf{f}_p^{3D}$, and only the 3D network and projection layers are trainable. All 2D components (MLLM + SAM) are frozen and serve solely as feature extractors. We reuse the SAM decoder within LISA to extract 2D binary masks while separately using LISA's reasoning head to obtain [SEG] tokens. This modular design decouples semantic reasoning and geometric supervision, making the framework compatible with other MLLMs or mask generators.

**Multi-modal Semantic Alignment.** Inspired by previous works [6, 5], we develop a token-for-Query mechanism to align multimodal semantics. We first establish point–pixel correspondences such that for a 3D point $p$ and view $i$, if point $p$ projects to pixel $(u, v)$ in the image and that pixel lies inside the mask ($M_i(u, v) = 1$), then view $i$ votes that point $p$ is part of the target segment. Conversely, if $p$ is visible in view $i$ but falls outside the mask, that view votes that $p$ is not part of the segment. After processing all views, each point $p$ accumulates multiple predictions from different views. We then aggregate these multi-view predictions using the attention weights. Importantly, along with these binary masks, we also associate each point with multi-modal features: for every view $i$ that observed point $p$, we retrieve the vector of features of the image $\mathbf{f}_{p,i}^{2D}$ from the SAM encoder in the corresponding pixel, and we carry the unified embedding $\mathbf{q}$ as a semantic token.

To align each 3D point with the unified embedding $\mathbf{q}$, we first map it using a learned linear projection due to differences in the dimensions of the modality. Let $\mathbf{t}$ denote the transformed unified query embeddings. Next, we compute semantic logits for each paired point taking the dot product between the transformed embedding and the 3D point features $s_p = \mathbf{f}_p^{3D\top} \cdot \mathbf{t}$ and the sample of corresponding predicted logits from the image-based segmentation predictions $\hat{m}_{p,i} = \hat{\mathbf{M}}_{i,u,v}$. Finally, we formulate the semantic alignment loss ($\mathcal{L}_{\text{MMS}}$) using binary cross-entropy (BCE) to minimize discrepancies between 3D logits and the sampled 2D predicted logits:

$$\mathcal{L}_{\text{MMS}} = \text{BCE}(s_p, \hat{m}_{p,i}) = -\frac{1}{|\mathcal{T}|} \sum_{p,i \in \mathcal{T}} [\hat{m}_{p,i} \log(\sigma(s_p)) + (1 - \hat{m}_{p,i}) \log(1 - \sigma(s_p))], \quad (1)$$

where $\sigma(\cdot)$ denotes the sigmoid activation function, and $\mathcal{T}$ represents the set of all paired points. This token-for-query-based alignment ensures that the same object (as described by the implicit text) yields a consistent latent representation across different views. Intuitively, minimizing $\mathcal{L}_{\text{MMS}}$ encourages the 3D model to produce consistent latent representations for the target object described implicitly by the textual query, thereby anchoring the 3D semantic features closely to the unified semantic embedding across all relevant views.

**Spatial Consistency Loss.** While $\mathcal{L}_{\text{MMS}}$ binds the network to the unified semantic token, we also enforce spatial consistency between 3D and 2D modalities. For each pseudo-labeled point $p$, recall that we have one or more associated image feature vectors $\mathbf{f}_{p,i}^{2D}$ from the views detected $p$. These features capture the appearance of the object from those viewpoints. We impose the 3D point feature $\mathbf{f}_p^{3D}$ to be close to the image features of the same point, which encourages the 3D model to agree with the 2D observations and maintain cross-view coherence. Concretely, we define a spatial consistency loss using cosine similarity. For each pair $(p, i)$ (point $p$ visible in view $i$ with feature $\mathbf{f}_{p,i}^{2D}$), we maximize the cosine similarity $\cos(\mathbf{f}_p^{3D}, \mathbf{f}_{p,i}^{2D})$. Equivalently, we minimize:

$$\mathcal{L}_{\text{spatial}} = \frac{1}{|\mathcal{T}|} \sum_{(p,i) \in \mathcal{T}} \left( 1 - \frac{\mathbf{f}_p^{3D} \cdot \mathbf{f}_{p,i}^{2D}}{\|\mathbf{f}_p^{3D}\| \|\mathbf{f}_{p,i}^{2D}\|} \right), \quad (2)$$

where $\mathcal{T}$ is the set of all point–view pairs for which point $p$ was labeled as target in view $i$. Minimizing $\mathcal{L}_{\text{spatial}}$ drives $\mathbf{f}_p^{3D}$ and $\mathbf{f}_{p,i}^{2D}$ in the same direction. We combine these objectives into the overall training

loss for the 3D network:

$$\mathcal{L}_{\text{total}} = \mathcal{L}_{\text{MMS}} + \lambda \mathcal{L}_{\text{spatial}}, \tag{3}$$

with a balancing coefficient $\lambda$.

## 3.3 Inference on 3D Scenes

At inference, the model takes as input a set of multi-view images of an unseen 3D scene and an implicit user instruction with the point cloud. Following the same pseudo-label generation pipeline, we compute a unified embedding $\mathbf{q}$ from the [SEG] tokens. The 3D model produces per-point features $\mathbf{f}_p^{3D}$, and cosine similarity between $\mathbf{f}_p^{3D}$ and $\mathbf{t}$ determines each point's probability of belonging to the queried object. Points exceeding a threshold are classified as part of the target, yielding the final 3D segmentation mask.

# 4 Experiments

In this section, we present the experimental results for three challenging tasks, focusing on 3D reasoning segmentation, intention grounding. For the *3D reasoning segmentation* task, we adopt Reason3D [23] (derived from ScanNet [12] and Matterport3D [3]) and Instruct3D [19] (derived from ScanNet++ v1 [64]) as benchmarks. Due to the limited open-source evaluation datasets in this area, we try to evaluate our method on similar tasks as 3D intention grounding (3D-IG) [32] and grounding without object names (VG-w/o-ON), which is an interesting benchmark introduced by [59]. These two datasets are built on ScanNet [12], taking advantage of its scene annotations for evaluation. Refer to Appendix A for more about the dataset and implementation details. Following SegPoint [19] and Reason3D [23], we evaluate in two stages: (1) Accuracy@kIoU (k=0.25/0.5) for target grounding correctness, and (2) mIoU for segmentation quality after successful grounding. This ensures that the evaluation reflects reasoning ability before mask quality.

## 4.1 Main Results

**3D Reasoning Segmentation.** As shown in Table 1, MLLM-For3D shows significant gains in 3D Reasoning Segmentation Precision over Reason3D on the Instruct3D benchmark [19]. Removing explicit instructions and annotations from Instruct3D makes the task much harder, since models can no longer rely on direct object names or step-by-step instructions and must infer the user's intent from implicit hints. In this setting, MLLM-For3D achieves about 15% higher Acc@0.25 and 10% higher Acc@0.50 than Reason3D. It also improves the mean IoU by 10 points, indicating more precise mask predictions. These improvements highlight MLLM-For3D's stronger reasoning capability: it can understand complex or implicit instructions to segment the correct 3D regions even when keywords are missing. In contrast, the Reason3D baseline struggles without explicit instructions, since it was designed to output masks based on more direct textual descriptions. In general, MLLM-For3D's ability to interpret implicit instructions leads to better performance on the 3D reasoning segmentation task. The "MLLM-based Model (w/ label)" serves as a reference baseline combining multi-view pseudo-labels and available 3D ground-truth masks. Both supervise the 3D network jointly under a hybrid training regime, using MinkowskiNet14 [10] as the backbone.

**3D Intention Grounding (3D-IG).** 3D Intention Grounding is a novel challenging task, which requires detecting the object that fulfills an implicit human intention. This setting is quite similar to what we have defined in the previous Section 1, but still ignores the spatial relations. IntentNet, a task-specific model, achieves 41.9% AP@0.25 and 25.4% AP@0.50 on this benchmark. MLLM-For3D surpasses this with 6–7 points higher AP@0.25 and 5 points higher AP@0.50, indicating that it more reliably identifies the correct object from only the implied intent.

For further comparison, shown in Table 2, our MLLM-For3D is evaluated under two settings: (i) with labeled 3D-text data (pink rows) and (ii) without labels (yellow rows). Despite the absence of manual 3D annotations in the label-free setting, our model surpasses the specialized IntentNet[32] by a notable margin. For example, comparing the best label-free variant VideoLISA (MLLM-For3D) with IntentNet, we observe an improvement of +9.0 points in AP@0.25 (41.90% $\rightarrow$ 50.89%) and +16.2 points in AP@0.50 (25.36% $\rightarrow$ 41.56%). This indicates that a multimodal LLM can better interpret various intention phrases and incorporate broader world knowledge than a system trained on fixed detection templates. Meanwhile, Reason3D [23] remains slightly ahead of our approach.

Table 1: **3D Reasoning Segmentation Results.** Comparison across ScanNet, Matterport3D, and ScanNet++ datasets. The evaluation metrics include accuracy at IoU thresholds 0.25, 0.50, and mean IoU. † denotes models fine-tuned using the filtered Instruct3D training set for fair comparison. [0] denotes a zero-shot experiment. The color gradient indicates different MLLM backbones and training conditions (w/ label or w/o label) for MLLM-For3D models.

| 3D Reasoning Segmentation Methods | Venue | Modality | Reason3D (ScanNet) | | | Reason3D (Matterport3D) | | | Instruct3D (ScanNet++) | | |
|---|---|---|---|---|---|---|---|---|---|---|---|
| | | | Acc@0.25 | Acc@0.50 | mIoU | Acc@0.25 | Acc@0.50 | mIoU | Acc@0.25 | Acc@0.50 | mIoU |
| *non-LLM-based Model (w/ label)* | | | | | | | | | | | |
| TGNN† [24] | [AAAI'22] | 3D | - | - | - | - | - | - | 4.76 | 4.76 | 3.51 |
| 3D-STMN† [58] | [AAAI'24] | 3D | 25.43 | 17.78 | 18.23 | 20.68 | 10.81 | 13.47 | - | - | - |
| Intent3D[0] [32] | [ICLR'25] | 3D | 29.12 | 19.26 | - | 19.70 | 12.83 | - | 9.71 | 3.20 | - |
| Intent3D† [32] | [ICLR'25] | 3D | 20.57 | 19.46 | - | 13.52 | 9.42 | - | 23.30 | 20.41 | - |
| *LLM-based Model (w/ label)* | | | | | | | | | | | |
| Segpoint [19] | [ECCV'24] | 3D | - | - | - | - | - | - | 23.7 | 15.6 | 17.2 |
| Reason3D [23]† | 3DV'25 | 3D | 43.21 | 32.10 | 31.20 | 31.22 | 17.43 | 19.54 | 18.35 | 10.55 | 12.43 |
| *MLLM-based Model (w/ label)* | | | | | | | | | | | |
| LISA-7B (MLLM-For3D) | - | 3D+2D | 45.53 | 39.54 | 31.94 | 38.31 | 31.46 | 22.56 | 45.50 | 38.70 | 31.90 |
| LISA-13B (MLLM-For3D) | - | 3D+2D | 47.32 | 39.92 | 31.44 | 39.42 | 32.43 | 23.20 | 46.70 | 39.00 | 32.30 |
| VideoLISA (MLLM-For3D) | - | 3D+2D | **48.45** | **41.02** | **39.82** | **39.81** | **29.40** | **24.97** | **48.20** | **40.40** | **34.50** |
| *MLLM-based Model (w/o label)* | | | | | | | | | | | |
| LISA-7B (MLLM-For3D) | - | 3D+2D | 39.38 | 31.27 | 30.19 | 30.10 | 22.04 | 20.71 | 39.10 | 29.70 | 23.90 |
| LISA-13B (MLLM-For3D) | - | 3D+2D | 40.92 | 33.40 | 32.10 | 31.68 | 23.33 | 20.78 | 40.90 | 30.50 | 26.40 |
| VideoLISA (MLLM-For3D) | - | 3D+2D | **44.18** | **34.80** | **32.90** | **33.41** | **27.99** | **22.50** | **41.00** | **32.10** | **28.20** |

Table 2: **Evaluation of 3D Intention Grounding on the Intent3D [32] validation set.** The best results are in **bold**, and the second-best results are underlined. † indicates that we re-trained Reason3D using the Intent3D training set for a fair comparison on this benchmark. [0] indicates a zero-shot setting.

| 3D Intention Grounding Methods | Venue | Modality | Language Backbone | Intent3D (val) | | | | |
|---|---|---|---|---|---|---|---|---|
| | | | | Acc@0.25 | Acc@0.50 | AP@0.25 | AP@0.50 | mIoU |
| *non-LLM-based Model* | | | | | | | | |
| BUTD-DETR [27] | [ECCV'22] | 3D | RoBERTa [41] | 47.12 | 24.56 | 31.05 | 13.05 | - |
| EDA [59] | [CVPR'23] | 3D | RoBERTa [41] | 43.11 | 18.91 | 14.02 | 5.00 | - |
| 3D-VisTA [69] | [ICCV'23] | 3D | - | 42.76 | 30.37 | 36.10 | 19.93 | - |
| IntentNet [32] | [ICLR'25] | 3D | RoBERTa [41] | 58.34 | 40.83 | 41.90 | 25.36 | - |
| *LLM-based Model* | | | | | | | | |
| Chat-3D-v2[0] [21] | [NIPS'24] | 3D+2D | Vicuna-7B [52] | 5.86 | 5.24 | 0.15 | 0.13 | - |
| Chat-Scene [21] | [NIPS'24] | 3D+2D | Vicuna-7B [52] | 36.71 | 32.78 | 3.23 | 2.58 | - |
| Reason3D† [23] | [3DV'25] | 3D | Flan-T5 [11] | **61.71** | **51.68** | - | - | **47.30** |
| *MLLM-based Model (w/ label)* | | | | | | | | |
| LISA-7B (MLLM-For3D) | - | 3D+2D | LLaVA-2 [40] | 57.31 | 47.92 | - | - | 42.98 |
| LISA-13B (MLLM-For3D) | - | 3D+2D | LLaVA-2 [40] | 58.40 | 48.75 | - | - | 44.13 |
| VideoLISA (MLLM-For3D) | - | 3D+2D | LLaVA-Phi-3-V [48] | 59.90 | 50.01 | - | - | 45.18 |
| *MLLM-based Model (w/o label)* | | | | | | | | |
| LISA-7B (MLLM-For3D) | - | 3D+2D | LLaVA-2 [40] | 48.24 | 39.61 | - | - | 34.53 |
| LISA-13B (MLLM-For3D) | - | 3D+2D | LLaVA-2 [40] | 49.92 | 40.10 | - | - | 35.75 |
| VideoLISA (MLLM-For3D) | - | 3D+2D | LLaVA-Phi-3-V [48] | 50.89 | 41.56 | - | - | 36.92 |

We attribute this gap to Reason3D's mask-as-embedding paradigm, which excels at implicit intent reasoning by fine-tuning LLM for direct segmentation tokens. However, MLLM-For3D still shows strong generalization in human intention reasoning, effectively combining detection and segmentation reasoning in a label-free manner. In contrast, IntentNet is primarily tailored for detection, and Reason3D focuses on search-plus-segmentation. The multimodal design of our method instead offers a more balanced approach, achieving second-best results while using no labels.

**VG-w/o-ON: Visual Grounding without Object Names** In the 3D visual grounding without object names task, MLLM-For3D achieves state-of-the-art results as shown in Table 3, outperforming the EDA baseline and others. VG-w/o-ON is a particularly challenging benchmark variant where the language query describes the spatial relationship of the target object without explicitly naming it. Conventional 3D referring models struggle here since they typically rely on matching object names in the query. In fact, we observe a drastic drop in baseline performance: methods such as ScanRefer [4] and TGNN [24] see their accuracy plunge to nearly chance level (e.g., 10% success) when object names are missing. Even EDA [59] reaches only 26. 5% Acc@0.25 and 21. 6% Acc@0.50 when the object names are missing, much lower than in normal queries. MLLM-For3D

Table 3: **Evaliation results on VG-w/o-ON (val) evaluated by Acc and mIoU.**; * auxiliary mask head. † indicates that we re-trained Reason3D using the ScanRefer[4] training set for a fair comparison on this benchmark.

| Methods | Venue | Acc@0.25 ↑ | Acc@0.50 ↑ | mIoU ↑ |
|---|---|---|---|---|
| *LLM-based Model (w/ label)* | | | | |
| ScanRefer [4] | [ECCV'20] | 10.51 | 6.20 | - |
| TGNN* [24] | [AAAI'21] | 11.64 | 9.51 | 8.13 |
| InstanceRefer [42] | [ICCV'21] | 13.92 | 11.47 | - |
| BUTD-DETR [27] | [ECCV'22] | 11.99 | 8.95 | - |
| M3DRef-CLIP [66] | [ICCV'23] | 18.3 | 14.8 | 10.29 |
| EDA [59] | [CVPR'23] | 26.50 | 21.20 | - |
| Reason3D [23] | [3DV'25] | 17.64 | 13.11 | 13.05 |
| IntentNet [32] | [ICLR'25] | 28.12 | 22.63 | 18.92 |
| *MLLM-based Model (w/ label)* | | | | |
| LISA-7B (MLLM-For3D)† | - | 31.88 | 29.90 | 28.10 |
| LISA-13B (MLLM-For3D) | - | 32.52 | 30.15 | 29.81 |
| VideoLISA (MLLM-For3D) | - | **33.12** | **31.21** | **30.45** |
| *MLLM-based Model (w/o label)* | | | | |
| LISA-7B (MLLM-For3D) | - | 26.49 | 22.12 | 21.05 |
| LISA-13B (MLLM-For3D) | - | 27.31 | 24.49 | 23.93 |
| VideoLISA (MLLM-For3D) | - | 29.50 | 25.61 | 24.28 |

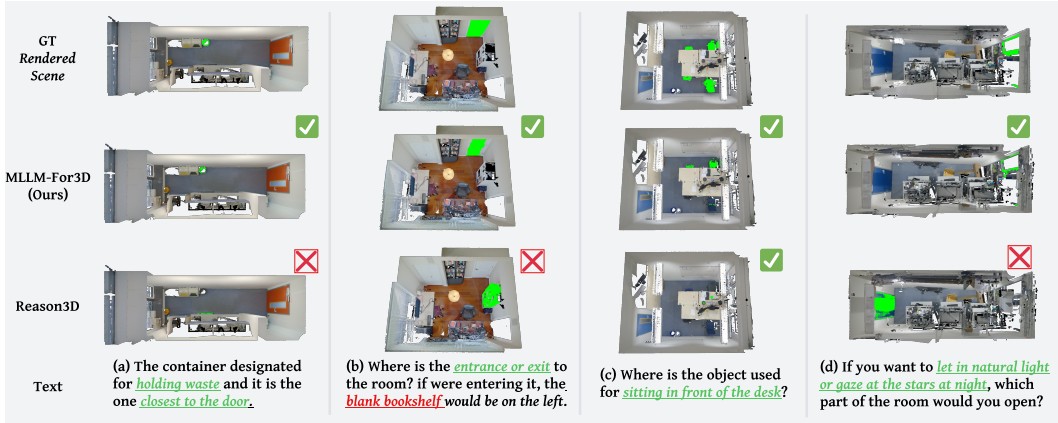

Figure 3: Visual comparisons of our MLLM-For3D versus a previous state-of-the-art method Reason3D on Intruct3D datasets. For each row, we show the ground-truth rendered scene (left), the baseline's prediction, our result, and the textual query. Our method accurately interprets implicit user instructions and produces coherent 3D masks.

overcomes this limitation by using the descriptive and contextual instructions of the query to infer the target. It delivers roughly 12–13% higher Acc@0.25 and 8–9% higher Acc@0.50 than EDA on the VG-w/o-ON benchmark, along with a notable improvement in mIoU.

Such results denote that our approach is capable of spatial reasoning by connecting the spatial relationship to the correct object in space. This contextual reasoning allows it to maintain high grounding accuracy despite the missing noun, whereas methods like EDA falter because they try to decompose the sentence and end up misled or unsure without an explicit object name. Using the abundant semantic information of a 2D MLLM, our method fills the semantic gaps (the missing object names) with informed guesses and uses the 3D visual input to confirm those guesses. This results in superior grounding performance under this no-name condition.

## 4.2 Visual Comparisons

Figure 3 presents qualitative examples comparing our **MLLM-For3D** framework against a previous state-of-the-art baseline on 3D reasoning segmentation tasks. Each row shows the ground-truth rendered scene, the predicted mask from the baseline, our result, and the text query. In the first example (left two columns), for the instruction 'The container designated to hold waste and it is the

Table 4: Ablation study evaluating the effectiveness of each proposed component and view configuration on the Instruct3D and VG-w/o-ON validation sets. Colored values indicate performance gain or drop compared to baseline (a).

| Ablation Target Setting | $+\mathcal{L}_{\text{MMS}}$ | $+\mathcal{L}_{\text{spatial}}$ | Instruct3D (val) | | | VG-w/o-ON (val) | | |
|---|---|---|---|---|---|---|---|---|
| | | | Acc@0.25 | Acc@0.50 | mIoU | Acc@0.25 | Acc@0.50 | mIoU |
| (a) Baseline (LISA-7B) | | | 29.25(ref) | 25.26(ref) | 19.75(ref) | 20.20(ref) | 17.24(ref) | 11.56(ref) |
| (b) w/o Token-for-Query | | ✓ | 33.92(+4.67) | 28.53(+3.27) | 20.93(+1.18) | 24.50(+4.30) | 20.70(+3.46) | 19.92(+8.36) |
| (c) w/o Spatial Consistency | ✓ | | 34.53(+5.28) | 28.75(+3.49) | 21.02(+1.27) | 23.20(+3.00) | 19.10(+1.86) | 18.48(+6.92) |
| (d) #View 2 | ✓ | ✓ | 33.21(+3.96) | 26.45(+1.19) | 21.30(+1.55) | 23.04(+2.84) | 20.13(+2.89) | 18.92(+7.36) |
| (e) #View 8 | ✓ | ✓ | 39.08(+9.83) | 29.10(+3.84) | 23.50(+3.75) | 25.22(+5.02) | 22.00(+4.76) | 20.71(+9.15) |
| (f) Full Config (LISA-7B) | ✓ | ✓ | 39.10(+9.85) | 29.70(+4.44) | 23.90(+4.15) | 26.49(+6.29) | 22.12(+4.88) | 21.05(+9.49) |
| (g) Full Config (LISA-13B) | ✓ | ✓ | **40.90**(+11.65) | **30.50**(+5.24) | **26.40**(+6.65) | **27.31**(+7.11) | **24.49**(+7.25) | **23.93**(+12.37) |

closest to the door', the baseline incorrectly merges multiple objects or fails to capture the target, whereas **MLLM-For3D** precisely identifies only the correct container. In another example (right columns), given 'If you want to let in natural light and fresh air once the air starts at night, which part of the room would you open?', the baseline misses key portions under occlusion, while our approach segments the relevant object more completely. These comparisons show that our model follows high-level instructions more faithfully, resolving common failure modes by leveraging LLM-driven 3D reasoning.

## 4.3 Ablation Studies & Analysis

We conduct extensive experiments to verify the contribution of each component and view design. The results are shown in Table 4.

**(a) 2D Projection Baseline**. We first apply a projection-only baseline where 2D masks from LISA are unprojected to 3D directly, without spatial reasoning. We unproject the multi-view masks generated by LISA-7B back to the point cloud. This naive approach yields a much lower accuracy, roughly 30% lower mIoU than our full 3D method. The projected baseline often produces incomplete or misaligned segmentations, since each view sees only part of the scene without cross-view consistency enforcement. This highlights the limited spatial reasoning ability of the existing 2D reasoning segmentation model.

**(b) w/o Token-for-Query**. Removing the token-guided semantic alignment leads to false positives. The model cannot consistently localize the queried object, activating multiple regions per query. With the token-for-query in place, the model focuses on one target at a time, reducing false positives by 30%. This mechanism ensures that one coherent mask per query and consistent segmentation is provided, even when multiple queries are issued in one scene.

**(c) w/o Spatial Consistency**. Disabling this module leads to inconsistent masks and a drop of 2-4% in segmentation accuracy. Enforcing spatial consistency across views improves performance: By aligning features in 3D space, the model learns a unified segmentation that is viewpoint-invariant, resolving ambiguities from single perspectives.

**(d-f) View Number Ablation**: We evaluate the impact of the number of views (2, 4, and 8) used during 2D inference. As shown in Table 4, **2 views** leads to lower accuracy, as many occluded or peripheral objects are invisible from sparse views, resulting in incomplete 3D supervision. **8 views** slightly improves over 2 views but falls short of the 4-view setup. Although it offers more contextual information, it also introduces redundant or conflicting signals from occluded perspectives. In practice, increasing the number of views from 4 to 8 doubles the inference cost without a consistent performance gain. We empirically found 4 views to balance accuracy and efficiency. Repeating 5 random 4-view configurations on the Instruct3D validation set yielded stable results (Acc@0.25: 39.10±0.61, Acc@0.5: 29.70±0.48, mIoU: 23.90±0.52), confirming robustness of our token attention mechanism to stochastic view sampling.

**(f,g) LISA-13B Backbone**: We also test a stronger MLLM (13B vs 7B). A larger model improves reasoning ability and segmentation precision, showing that the architecture is scalable. Our best setup includes both modules, a balanced number of views, and a LISA-13B backbone.

**Additional Comparison with 3D-MLLMs.** To further validate the effectiveness of our label-free design, we additionally compare MLLM-For3D with representative 3D-MLLM pipelines that combine large language models with 3D segmentation backbones, including ChatScene [55, 55] and Mask3D [50]. As detailed in Appendix B, even when ChatScene is fine-tuned on Intent3D or combined with Mask3D for 3D-SAM inference, our zero-shot MLLM-For3D achieves substantially higher accuracy and mIoU without any 3D supervision. Specifically, it surpasses fine-tuned ChatScene+Mask3D by 21.7% Acc@0.25 and 14.9% Acc@0.50 on Intent3D, and outperforms the zero-shot ChatScene baseline by over 30% Acc@0.25 and 25% Acc@0.50 on Reason3D. These consistent improvements demonstrate that MLLM-For3D can transfer semantic and reasoning knowledge from 2D MLLMs more effectively than current 3D-MLLM architectures, achieving competitive or superior performance without costly fine-tuning or labeled 3D data.

## 5 Conclusion & Limitations

We introduce **MLLM-For3D**, a novel framework that adapts MLLM for 3D reasoning segmentation using a *label-free* paradigm. Our approach tackles challenges such as single-view hallucination and cross-view inconsistencies by employing an attention-based fusion strategy alongside a token-for-Query mechanism, enabling coherent multi-view pseudo-label generation without any annotations. Experiments on three challenging benchmarks reveal that our method achieves SOTA performance in label-free settings and demonstrates further improvements when 3D labels are made available. These results confirm the effectiveness of our adaptation strategy and open new avenues for scalable, language-guided 3D scene understanding. However, the method can be computationally demanding as it involves multiple inferences of 2D MLLMs. Future work may explore more efficient architectures, better uncertainty modeling of pseudo-labels, and broader generalization to complex, real-world 3D environments.

## Acknowledgments and Disclosure of Funding

Mingming Gong was supported by ARC DP240102088 and WIS-MBZUAI 142571.

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

# A  Technical Appendices and Supplementary Material

## A.1  Datasets Descriptions

We base our 3D reasoning-segmentation experiments on existing indoor scene benchmarks with generated language queries. The Reason3D dataset [23] provides query-conditioned object masks on ScanNetV2 [12] and Matterport3D [3]. Each sample in Reason3D is a single-room point cloud paired with a natural-language query and a binary mask of the target object. We follow the official splits and statistics: Matterport3D contributes 934 training and 837 validation samples, while ScanNetV2 contributes 405 training and 308 validation samples. These datasets supply (implicit) query–object-ID supervision for reasoning segmentation.

We use Instruct3D [19], a harder benchmark derived from the high-fidelity ScanNet++ v1 [64] dataset. Instruct3D omits explicit object names or step-by-step instructions, requiring models to infer both

user intent and spatial relationships from context. The filtered Instruct3D split contains 136 training scenes and 45 validation scenes, yielding 1,034 and 321 query-answer (QA) pairs, respectively. These QA pairs consist of implicit queries with spatial relationships and segmentation masks of the referred objects. Removing direct mentions of object names and requiring spatial cues makes Instruct3D significantly more challenging.

Finally, we evaluate grounding on two 3D spatial-reasoning datasets built on ScanNet scenes: 3D-IG (Intent3D) [32] and VG–w/o–ON [59]. The Intent3D dataset (3D intention grounding) contains free-form human-intent descriptions paired with target object detections, while VG–w/o–ON ("visual grounding without object names") contains spatial queries that deliberately omit object names. Both are constructed atop ScanNet annotations and test the model's ability to localize objects from descriptive, context-dependent language.

## A.2 Implementation Details

Our 3D segmentation network is implemented using MinkowskiNet14 as the backbone, built on the PyTorch framework. Throughout training, both the multimodal large language model (MLLM, specifically LISA-7B) and the Segment-Anything Model (SAM) remain frozen to leverage pre-trained 2D multimodal knowledge, while only the 3D projection model is optimized. Training is performed on four NVIDIA A100 GPUs (40 GB each).

We summarize key training hyperparameters and configurations used across the ScanNetV2, Matterport3D, and ScanNet++ v1 datasets in Table 5. For optimization, we employ stochastic gradient descent (SGD) with momentum set to 0.9 and a weight decay of $1 \times 10^{-4}$. Data augmentations such as random rotation around the upright axis, random flips on point clouds, and random horizontal flips and resized crops on images were consistently applied to enhance model generalization.

Table 5: Training configurations across different datasets.

| Dataset | Backbone | Batch Size | LR | Epochs | GPUs | Voxel Size | Max Sweeps |
|---|---|---|---|---|---|---|---|
| ScanNetV2 | MinkowskiNet14 | 8 | 0.10 | 40 | 4 | 0.05 m | 1 |
| Matterport3D | MinkowskiNet14 | 4 | 0.10 | 40 | 4 | 0.05 m | 1 |
| ScanNet++ (Instruct3D) | MinkowskiNet14 | 4 | 0.01 | 40 | 4 | 0.05 m | 1 |

Training times vary depending on the complexity and size of the data set, especially the language number. ScanNetV2 and Matterport3D (Reason3D) typically converges within approximately 20 hours, whereas ScanNet++ v1 (Instruct3D) datasets require roughly 30–40 hours. These configurations were empirically chosen to ensure robust convergence across all datasets. Batch size and GPUs are configured based on a PyTorch Lightning DataModule, with batch size set by dividing the total batch size by the number of GPUs (i.e., `self.batch_size = config["batch_size"] // config["num_gpus"]`)

## A.3 Model Architecture

Our model comprises two fused branches: a 3D sparse-CNN branch for processing voxelized scene data, and a 2D vision-language branch for language-conditioned image segmentation. The 3D branch is implemented as a UNet-style sparse convolutional network (MinkowskiNet14). All 3D convolutions use a $3 \times 3 \times 3$ kernel and produce 512-dimensional features per occupied voxel. Batch-normalization layers (with momentum 0.05) follow each convolution. The network follows an encoder–decoder ("U-Net") design with skip connections between corresponding scales. During training, only this 3D branch is updated; the 2D branch parameters remain fixed.

**3D Sparse Convolutional Branch.** We adopt the Minkowski Engine's 3D U-Net backbone ("MinkowskiNet14"). Following prior practice in high-dimensional CNNs, all convolutions use kernel size $3 \times 3 \times 3$, with hyper-cross patterns ("+" indicates 1-D in the third dimension). The network has multiple down- and up-sampling stages (forming a U-shape) with symmetric skip connections. Each sparse conv layer produces 512 features per voxel (after the final encoder stage), and we attach a BatchNorm (momentum 0.05) and ReLU nonlinearity to each layer. The output of the 3D branch is a set of per-voxel features over the scene. These features will be aligned (during training) to

the projected 2D segmentations. All weights in this 3D branch are learnable, while the 2D branch is kept frozen.

**2D Vision-Language Segmentation Branch**. The 2D branch leverages pretrained LISA multimodal LLMs [36, 2] to perform language-guided image segmentation. We use the publicly released LISA-7B, LISA-13B, and VideoLISA models as frozen multimodal feature extractors. In all cases, we operate in single-frame inference mode (treating VideoLISA as an image model), with no temporal aggregation. The LISA models consist of a CLIP ViT-L/14 vision encoder and a LLaVA-based language model. The vision encoder extracts dense image features from each input frame, and the decoder (a lightweight dilated convolutional network) upsamples the mask logits to the original image resolution. The entire LISA model, including the SAM-style segmentation decoder, remains frozen during training. To guide the segmentation process, we use LISA's vocabulary, which includes a special [SEG] token that acts as a semantic anchor. Given a natural language instruction, we prepend the [SEG] token to the input prompt. After multimodal processing by the LLM, the model generates an embedding corresponding to [SEG], denoted . This embedding is projected via a learned linear transformation to match the dimensionality required by the SAM decoder:

$$\mathbf{h}_{\text{seg}} = W \tilde{\mathbf{h}}_{\text{seg}}, \tag{4}$$

where $W$ is a learnable projection matrix.

**SAM Decoder.** The projected token embedding [SEG] is used to prompt the frozen SAM decoder. The decoder processes the image features and the token prompt to produce a coarse segmentation mask. This mask is subsequently refined and upsampled by a lightweight dilated convolutional decoder, producing the final high-resolution binary segmentation mask.

**Frozen Inference** We do not modify the architecture or parameters of LISA or SAM. All 2D weights are frozen, and only the 3D segmentation model is updated during training. This design enables effective language-conditioned segmentation with no additional finetuning of the 2D backbone, allowing the 3D model to inherit multimodal reasoning capabilities from LISA through differentiable pseudo-label projection.

# B  Additional Comparisons with 3D-MLLM Baselines

In this section, we provide additional quantitative comparisons with existing 3D-MLLM pipelines that combine large language models with 3D segmentation backbones, such as ChatScene [55, 21] and Mask3D [50].

## B.1  Evaluation on Intent3D and Reason3D

We evaluate both **3D Intention Grounding** and **3D Reasoning Segmentation** tasks using the public validation sets of Intent3D [32] and Reason3D [23]. For fair comparison, we selected Mask3D [50] as the 3D-SAM baseline, as it provides a pre-trained checkpoint on the ScanNet validation set and avoids dataset discrepancies between benchmarks.

For Intent3D, we follow the pipeline of ChatScene [21], where object ID strings are generated via a fine-tuned Vicuna-7B model and aligned with Mask3D object proposals to obtain the final 3D masks. For Reason3D, we evaluate zero-shot reasoning segmentation performance under the same experimental setup.

**1. 3D Intention Grounding on Intent3D**

| Model | Acc@0.25 | Acc@0.5 | mIoU |
|---|---|---|---|
| ChatScene (FT) + Mask3D | 36.71 | 21.50 | 12.08 |
| **Ours (Zero-Shot)** | **58.40** | **41.00** | **26.15** |

**2. 3D Reasoning Segmentation on Reason3D**

| Model | Acc@0.25 | Acc@0.5 | mIoU |
|---|---|---|---|
| ChatScene (Zero-Shot) + Mask3D | 9.23 | 8.89 | 1.01 |
| **Ours (Zero-Shot)** | **44.18** | **34.80** | **32.90** |

Overall, our zero-shot pipeline consistently surpasses both fine-tuned 3D-MLLM + 3D-SAM frameworks on Intent3D and zero-shot settings on Reason3D, demonstrating strong generalization without any 3D supervision. The inferior performance of existing 3D-MLLMs mainly stems from two factors: (1) the scarcity of 3D training data for aligning visual features within the LLM embedding space, and (2) hallucinations and false positives produced by LLM responses. In contrast, **MLLM-For3D** leverages frozen 2D MLLMs and semantic alignment to achieve superior reasoning-based segmentation without costly fine-tuning or labeled 3D annotations.

