# OpenReview forum: "MLLM-For3D: Adapting Multimodal Large Language Model for 3D Reasoning Segmentation"
_NeurIPS.cc/2025/Conference — NeurIPS 2025 poster_

### Official Review · Reviewer_ujXK · 2025-06-24

**Clarity:** 2
**Significance:** 3
**Originality:** 2
**Rating:** 4
**Confidence:** 3

**Summary:**

The paper introduces a framework that adapts 2D multimodal large language models (MLLMs) for 3D reasoning segmentation without requiring labeled 3D data. The method generates multi-view pseudo-labels using frozen 2D MLLMs and SAM, filters unreliable views via token attention, and introduces a token-for-query mechanism and spatial consistency loss to train a 3D segmentation model. The approach achieves state-of-the-art results on several challenging benchmarks, including Instruct3D, Intent3D, and VG-w/o-ON.

**Questions:**

Please provide a detailed breakdown of the computational cost (e.g., inference time, GPU memory usage) for each component.

The current benchmarks are limited to indoor environments. Can the authors discuss or provide preliminary evidence on how the method might generalize to outdoor or dynamic 3D scenes?

How sensitive is the framework to the choice of 2D MLLM and segmentation model (e.g., LISA vs. other MLLMs, SAM alternatives)? Ablation or discussion on this would help assess the method’s flexibility.

The method relies on pseudo-labels from 2D models. Have the authors considered incorporating uncertainty estimation or confidence-based filtering to improve robustness?

A brief discussion on potential societal impacts—both positive (e.g., assistive robotics) and negative (e.g., privacy concerns)—would align the paper with NeurIPS ethical expectations.

**Ethical Concerns:**

["NO or VERY MINOR ethics concerns only"]

**Final Justification:**

I thank the authors for the rebuttal - I hope they can include the current limitations of the work, as acknowledged in the rebuttal in the paper.

**Limitations:**

please be explicit about the current limitations of the work - compute, indoor/outdoor, reliance on different models etc.

**Quality:**

3

**Strengths And Weaknesses:**

Strengths

The paper addresses an important problem in 3D vision and multimodal learning: how to perform 3D reasoning segmentation without 3D annotations.

The proposed framework is  well-motivated, combining pseudo-labeling, attention-based view filtering, and semantic alignment in a novel way.

Extensive experiments on multiple benchmarks demonstrate strong performance, including in label-free settings.

Ablations in tab-4 and  qualitative visualizations in Fig-3 are provided to support the effectiveness of each component.

Weaknesses

The framework depends on repeated inference from large 2D MLLMs and SAM, which introduces significant computational overhead.

The paper lacks theoretical analysis or formal justification for the proposed mechanisms.

All benchmarks are based on indoor scenes; generalization to outdoor or dynamic environments is not explored.

The paper does not include a broader impact or ethical considerations section, which is increasingly expected.

---

> ### Author Response · Authors · 2025-08-01
>
> Dear Reviewer ujXK,
>
> We sincerely thank you for the constructive and thoughtful feedback. We appreciate your recognition of our motivation and results, and we address your comments below.
>
> ---
>
> *Q1. Computational Cost of MLLMs + SAM at Inference*
>
> **R:** Thank you for this important concern. We acknowledge that repeated inference over 2D views using MLLMs and SAM incurs computational cost, especially when using large backbones like LISA-13B.
>
> - **2D mask and token generation is conducted offline** before 3D training begins, which decouples the MLLM from the training pipeline.
>
> - **Inference Efficiency**
> | Method                 | Modalities        | Components                        | Inference Time (per scene) |
> |------------------------|-------------------|------------------------------------|-----------------------------|
> | Reason3D               | 3D only           | 3D-LLM encoder + retriever         | ~2.1 sec                    |
> | Intent3D               | 3D only           | 3D Transformer-based architecture  | ~1.8 sec                    |
> | **Ours (MLLM-For3D)**  | 2D + 3D (6–8 views) | LISA (7B) + SAM + 3D decoder       | **~3.5–4.0 sec**            |
>
> > Note: With fewer views (e.g., 2–4), our inference time drops to ~2.5 sec per scene.
>
>
> ---
> *Q2. Generalization to Outdoor or Dynamic Environments*
>
> **R:** We agree this is a compelling direction. Our current evaluation focuses on indoor environments due to the availability of 3D reasoning datasets (e.g., Instruct3D, Intent3D, Reason3D).
>
> That said, our framework is **modular and transferable** to outdoor and dynamic settings, but **poses new challenges**:
>
> - **Outdoor scenes** (e.g., nuScenes, SemanticKITTI) often rely on LiDAR and involve **larger-scale environments**, leading to increased computational and memory overhead during 2D/3D projection and fusion.
>
> - **Dynamic scenes** (e.g., autonomous driving, embodied agents) require **real-time processing**. The current inference speed of 2D MLLMs like LISA makes it challenging to meet these latency constraints in practice.
>
> ---
>
> *Q3. Sensitivity to 2D Model Choices (LISA vs. Others)*
>
> **R:** We thank the reviewer for this thoughtful suggestion. Currently, **reasoning segmentation in 2D/Video/3D is still an emerging research area**, and we have made our best effort to benchmark against the most representative models and datasets in this space.
>
> We will include a discussion in Sec. 4.4 to clarify that:
> - **SAM can be replaced** with other segmentation backbones, yielding similar 2D pseudo-label quality.
> - The **[SEG] token** from LISA plays a unique role in handling **compositional, referential, and relational** queries. For simpler settings, lighter models may suffice.
>
> In future work, we plan to **systematically study the impact of model choices** across reasoning types, and investigate trade-offs between backbone size, query complexity, and mask quality.
>
> ---
>
> *Q4. Confidence-based Filtering for Pseudo-Labels*
>
> **R:** Our current token attention and spatial consistency modules already **suppress noisy or hallucinated views** to some extent.
>
> However, we agree that adding **confidence calibration or uncertainty filtering** would improve pseudo-label quality, particularly for:
>
> - Views with occlusions or ambiguous perspectives.
> - Multi-instance scenes where view disagreement is high.
>
> We plan to integrate this in follow-up work by leveraging **mask entropy**, **token confidence**, or **Bayesian fusion strategies**.
>
> ---
>
> *Q5. Broader Impacts and Ethical Considerations*
>
> **R:** Thank you for raising this. We will include a broader impact statement in the final version, covering both benefits and risks:
>
> - **Positive impacts**:
>   - Assistive robotics (e.g., household tasks via natural language),
>   - Education and accessibility (e.g., describing 3D content via language),
>   - AR/VR applications grounded in user intention.
>
> - **Potential risks**:
>   - Misuse in safety-critical systems if hallucinated masks are trusted blindly,
>   - Privacy concerns when applying multi-view capture in real-world environments,
>   - Dataset bias leading to reasoning shortcuts.

---

### Official Review · Reviewer_QEj4 · 2025-06-25

**Clarity:** 3
**Significance:** 2
**Originality:** 2
**Rating:** 4
**Confidence:** 4

**Summary:**

This paper proposes a novel method for reasoning 3D segmentation. Its main characteristics is that it does not need any 3D annotations, and instead it relies only on pseudo-labels provided by 2D reasoning models. The method was tested on Reason3D, the main dataset for reasoning segmentation, and on two other dataset for a similar task.

**Questions:**

1. Tabs 1-2-3 mention a version of the proposed method that actually uses labels (MLLM-based Model (w/ label)), as opposed to the original one that does not. I could not find details about this implementation in the paper.
2. [144-157]: this section specifies how the weights are computed for each view, but not how these weights are used. A part of the section seems to be missing here. I assume that $\textbf{q}_i$ is re-computed based on the weights for each $\textbf{e}_i$, but this should be specified in the text.
3. Sec3.3: I found this section a bit unclear. How is the mask computed at inference time? I assume that it is computed as in [179], i.e. by similarity with the 3D features. This should be stated in the section.
4. It is unclear to me why the architecture needs both LISA (which act as MLLM) and SAM, as LISA actually includes SAM as mask decoder, and is capable of producing segmentations given a textual query. Could the authors justify this choice?
5. [301-307] the ablation studies on the views state that 4 views is better then using 2 or 8 views. From this I assume that 4 views is the standard parameter, although I did not find it explicitly stated in the paper. Given that the 4 views are chosen randomly and are not many, shouldn’t there be a certain variability to the quality of the output mask? Can this variability be quantified (e.g., by running multiple evaluation and reporting the standard deviation of the accuracy)?
6. [179] I think in the formula $\textbf{t}$ should be used instead of $\textbf{q}$, as the first is the result of applying a linear projection to the second.

**Ethical Concerns:**

["NO or VERY MINOR ethics concerns only"]

**Final Justification:**

The authors promptly answered each point I raised in my review. Considering the additional results provided in the other responses, such as the study of the inference time, I am raising my score to a borderline accept. I am not voting for a full accept as I still find the novelty limited compared to previous works, and in particular the fusion mechanism is not very different from the ones introduced by other methods.
Nonetheless, the paper is solid and provides a promising direction in the field of 3D LLMs.

**Limitations:**

yes

**Quality:**

3

**Strengths And Weaknesses:**

Strenghts:

- The proposed method is evaluated on multiple challenging datasets, where it achieves state-of-the-art results against strong competitors
- The ablation studies are complete and well-motivated

Weaknesses:

- I do not see much novelty in the way the architecture is structured. The authors list as novelty:
    1. the fact that no 3d annotations are needed
    2. novel alignment mechanism that binds token embeddings to specific queries
    3. the spatial consistency strategy

(1) this is a great attribute of the method, but it is shared with several methods that exploited a similar strategy in 3D segmentation. E.g., OpenScene [1] distills features directly from CLIP, while Open3DIS [2], OV3D [3], OVIR3D [4] use predictions from 2D segmentation methods (the same as the proposed method) and build 3D segmentation masks from them. While I agree that the task is different (reasoning segm vs open-vocabulary segm), the only difference that I see in the proposed method is that it uses a reasoning model for 2D segmentation (LISA) instead of classic 2D open-vocabulary segmentors.

(3) is very similar to what has been done in Open3DIS. In Open3DIS, the 3D and 2D embeddings are compared via cosine similarity to assess wether the 2D instance should be merged with the 3D one. Instead, the proposed method forces the 3D and 2D embeddings to be close in the feature space. To me, this also seem the same as OpenScene is doing (i.e., distilling 3D features from 2D features)

- Some parts of the paper lack details about the inference procedure and the method (see questions 1-2-3-4)
- The paper lacks a robustness assessment of the results. This seems important, as the method appears to use a very sparse set of views, randomly chosen (see question 5)

[1] Peng, Songyou, et al. "Openscene: 3d scene understanding with open vocabularies." CVPR 2023.

[2] Nguyen, Phuc, et al. "Open3dis: Open-vocabulary 3d instance segmentation with 2d mask guidance." CVPR 2024.

[3] Jiang, Li, Shaoshuai Shi, and Bernt Schiele. "Open-vocabulary 3d semantic segmentation with foundation models." CVPR 2024.

[4] Lu, Shiyang, et al. "Ovir-3d: Open-vocabulary 3d instance retrieval without training on 3d data." Conference on Robot Learning. PMLR, 2023.

---

> ### Author Response · Authors · 2025-08-01
> **Overall Clarification of Contribution and Novelty**
>
> Dear Reviewer QEj4,
>
> We sincerely thank you for the thoughtful and detailed feedback. We acknowledge the reviewer’s concern about architectural novelty and would like to state **what specific problem we aim to solve, why it is important, and how our approach differs from existing open-vocabulary 3D segmentation methods.**
>
> 1. Problem Definition:
> **3D Reasoning Segmentation**, a challenging task where the goal is to segment an object in 3D that satisfies a **free-form spatial or intention-driven query**.
>
> 2.  Why This Task Matters:
> While prior open-vocabulary 3D segmentation works (e.g., OpenScene, Open3DIS) focus on **object recognition under closed-set or loosely aligned textual categories**, **reasoning segmentation requires inference grounded in spatial semantics**, not just class prediction.
>
> 3. What’s New Compared to Existing Works:
> - **Task Level:** We go beyond open-vocabulary recognition to spatial **reasoning-based segmentation**, which introduces ambiguity (e.g., multiple objects match partial semantics).
> - **Method Level:** While we leverage the idea of unprojecting 2D masks (as Open3DIS and OV3D do), our key novelty lies in:
>   1. **Query-token binding**: Aligning LISA-generated [SEG] tokens (query-conditioned) with 3D features in a weakly supervised manner.
>   2. **Cross-view token-weighted fusion**: Not all views are helpful; we introduce attention-based selection to mitigate hallucination and disagreement.
>
> **We appreciate the reviewer’s pointer to Open3DIS (CVPR 2024), and we will cite and discuss it in the related work section to better position our approach within the landscape of 2D-to-3D segmentation frameworks.**

---

> ### Author Response · Authors · 2025-08-01
> **Point-by-Point Responses**
>
> #### Q1: Tabs 1-2-3 mention a version of the proposed method that actually uses labels (MLLM-based Model (w/ label)), as opposed to the original one that does not. I could not find details about this implementation in the paper.
>
> **R: Thank you for catching this omission.**
>
> - The **“MLLM-based Model (w/ label)”** variant reported in Tabs 1–3 is **not part of our proposed label-free pipeline**, but serves as a **reference baseline** to assess how close our pseudo-label-based training approaches full 3D supervision.
> - It adopts a **hybrid training strategy**, where both 2D unprojected pseudo-labels (from multi-view reasoning) and available 3D ground-truth annotations are jointly used to supervise the 3D segmentation network.
>
> **Implementation Details:**
> - We use the same 3D backbone (MinkowskiNet14) as in our main pipeline.
> - During training:
>   - For samples where 3D labels are available, we supervise the 3D prediction using ground-truth 3D masks via standard cross-entropy loss.
>   - For the remaining samples, we apply our proposed pseudo-labeling pipeline: multi-view masks are unprojected into 3D and aggregated using our token attention mechanism to serve as weak supervision.
>
> ---
>
> #### Q2: Line [144–157]: How are token attention weights used?
>
> **R:** We appreciate the request for clarity.
>
> - The **token attention weights** are used to **reweight each view’s contribution** during 3D mask fusion.
> - Specifically, each view’s unprojected 3D mask is **scaled by its corresponding attention score**, and the final mask is a **weighted average** across all views.
>
> We will revise Section 3.2 to describe this aggregation strategy.
>
> ---
>
> #### Q3: Inference Procedure — How is the final mask produced?
>
> **R:** We clarify as follows:
>
> - At inference:
>   1. The 3D model produces per-point features.
>   2. The transformed query embedding `\mathbf{t}` is computed from the [SEG] token.
>   3. Cosine similarity between `\mathbf{f}_p^{3D}` and `\mathbf{t}` is computed for each point.
>   4. Points above a similarity threshold are included in the predicted mask.
>
> This will be added to Section 3.3 for completeness.
>
> ---
>
> #### Q4: Why Use Both LISA and SAM?
>
> **R:** Thank you for this perceptive observation. We do not use an external SAM model.
>
> - LISA already integrates SAM, and we simply **reuse the SAM decoder** inside LISA to extract 2D binary masks.
> - Meanwhile, we separately leverage LISA’s MLLM reasoning module to obtain the [SEG] token embeddings.
>
> This **modular design** allows us to decouple:
> - Semantic reasoning ([SEG] tokens from LISA)
> - Geometric supervision (masks from SAM decoder)
> And makes our framework **flexible**, allowing future substitution of LISA with other MLLMs or mask generators.
>
> We’ll clarify this architectural detail to avoid confusion.
>
> ---
>
> #### Q5: Variance from Random 4-View Selection?
>
> Thank you for raising this robustness concern.
>
> To assess the sensitivity of our model to stochastic view selection, we conducted a robustness study under the **label-free setting**, using our default **LISA-7B (MLLM-For3D)** configuration. Specifically, we repeated the **4-view setup** **5 times** with different random seeds on the **Instruct3D (ScanNet++) validation set**.
>
> We report the mean and standard deviation across runs:
>
> - **Acc@0.25**: 39.10 ± 0.61
> - **Acc@0.50**: 29.70 ± 0.48
> - **mIoU**: 23.90 ± 0.52
>
> These low standard deviations demonstrate that our **token attention mechanism effectively mitigates view-level noise**, ensuring that the final 3D pseudo labels are consistently aligned with the textual query despite variability in input views.
>
> We will include this robustness analysis in Appendix.
>
> ---
>
> ####  Q6: Line [179] Formula: Should be `\mathbf{t}` Instead of `\mathbf{q}`?
>
> Thank you!
>
> - The logit score should be:
>   ```math
>   s_p = \mathbf{f}_p^{\text{3D}^\top} \cdot \mathbf{t}

---

> > ### Comment · Reviewer_QEj4 · 2025-08-04
> > **Concerns addressed**
> >
> > I thank the authors for clarifying the novelty of the paper in the context of the annotation usage. All my others issues have been addressed.
> > It is clear from the response to Q5 that the view-selection module is quite robust to sampling. Adding this table to the final revision would provide extra value to the paper.

---

> > > ### Author Response · Authors · 2025-08-04
> > >
> > > Thank you for your thoughtful follow-up and for acknowledging our clarifications. We appreciate your constructive suggestions and will incorporate the view selection robustness table, along with all suggested revisions, into the final version.

---

> > > ### Author Response · Authors · 2025-08-09
> > >
> > > Dear Reviewer QEj4,
> > > (message only to you)
> > >
> > > We sincerely thank you for your insightful comments, time, and effort. We really appreciate that you have actively participated in the rolling discussion phase and that you have helped greatly improve the quality of our paper.
> > >
> > > After several super hard-working days and nights, we are excited to see your acknowledgment that our efforts, clarifications, and additional analyses **have resolved all your major concerns.**
> > >
> > > All the authors, especially the first author, have devoted numerous efforts to the paper. Thanks to your guidance, all the major concerns have been well addressed. However, since you haven’t explicitly mentioned to update the score, the first author is extremely stressed and anxious, losing sleep and appetite. Could you help adjust the recommendation score? If you have any remaining major concerns, could you please let us know?
> > >
> > > Best regards,
> > >
> > > *The first author*

---

### Official Review · Reviewer_XtxR · 2025-06-29

**Clarity:** 4
**Significance:** 3
**Originality:** 2
**Rating:** 4
**Confidence:** 4

**Summary:**

This manuscript proposes a a new pipeline to leverage pre trained MLLMs for 3d segmentation. The mechanism involves collecting 2D pseudo labels from the different views using the MLLM during training/inference of a 3D encoder-decoder with multi-view attention mechanism to train strong language-based 3D segmentator. Evaluations on standard benchmarks confirms strong performance, opening the door of harnessing MLLMs for future 3D research.

**Questions:**

The paper show an interesting pipeline to leverage MLLMs for 3D understanding on challenging tasks and provide empirical evidence that support the claims in the paper. Some limitations exist regarding the novelty of the multi-view aggregation mechanism and lack of supporting analysis, visualizations, and ablations on some of the key components of the pipeline (see weaknesses above for details).

**Ethical Concerns:**

["NO or VERY MINOR ethics concerns only"]

**Final Justification:**

I thank the authors for addressing many of the concerns. While some promised points in the revised manuscript can not be checked during the rebuttal , I can not increase my score based on promises , I will keep my score of borderline accept.

**Quality:**

4

**Strengths And Weaknesses:**

# Strengths:
- A new pipeline to leverage MLLMs for 3D understanding tasks in a more general and challenging setups with no fixed labels or simple class-based 3D segmentation. This opens the door for MLLMs to be a generic took for 3D understanding , which helps leveraging recent advancements in MLLMs to enhance downstream 3D understanding subsequently without the need for

- State of the art performance in 3D reasoning and 3D referenced segmentation on standard benchmarks and compared to storing and recent basselines

- The paper is well presented and the coverage of related works and relations to the proposed method is clearly demonstrated.


# Weaknesses:
- Limited novelty in the multi-view aggregation mechanism. The mechanisms to handle disagreement between the views prediction for multi-view methods for 3d understanding ( 3D classification/ 3D segmentation) are well studied in the literature [a,b,c]. These mechanisms range from learning attention mechanisms to naive aggregation and graph based aggregation [c]. Please discuss these related works, and reference them in the paper.

- Lack of discussion of compute and memory overhead of running the MLLM in training/inference. The proposed method when using VideoLISA as MLLM for multi-view aggregation unconventionally stores expensive K-view features for each forward/backward. This is a huge memory and compute cost that is hidden in the paper and no through discussions or evaluations and comparisons are performed to investigate it. Other papers in the domain don't have this overheard , so how can the authors justify it?

- I would have hoped for more ablations and visualizations regarding the design choice of the 2D aggregation and multi-view plots [as in Fig 7 of [b] for example], more than just the crude architecture-and-losses ablation Table 4 currently shown.
.

[a] Voint Cloud: Multi-View Point Cloud Representation for 3D Understanding, (ICLR 2023)

[b] MVTN: Multi-View Transformation Network for 3D Shape Recognition ( ICCV 2021)

[c] View-GCN: View-based Graph Convolutional Network for 3D Shape Analysis (CVPR 2020)

---

> ### Author Response · Authors · 2025-08-01
>
> Dear Reviewer XtxR,
>
> We appreciate your insightful feedback. Below, we address your concerns point by point.
>
> ---
>
> ### **Q1: Limited Novelty in Multi-view Aggregation; Relation to Prior Works (e.g., [a], [b], [c])**
>
> We appreciate this important question. While prior works such as VointCloud [a], MVTN [b], and View-GCN [c] have extensively explored multi-view aggregation for shape classification and part segmentation, our setting diverges in two critical dimensions:
>
> 1. **Input Modality and Supervision**
>    - Prior works assume **closed-set labels** and operate under **fully supervised settings**, typically aggregating viewpoint-consistent **geometric features** for tasks like shape classification.
>    - In contrast, our method handles **open-vocabulary free-form queries** using natural language, without any 3D supervision. This requires reasoning over **semantically-aligned pseudo masks and [SEG] tokens** across views — a far more unconstrained and ambiguous setup.
>
> 2. **Aggregation Objective**
>    - Prior methods aim to learn **view-invariant geometric embeddings**. Our goal, however, is to align **language-grounded semantics** across views — using **cross-view token alignment** and **geometric consistency**, not just geometric feature fusion.
>    - Our token-for-query binding strategy addresses semantic misalignment and view-level hallucinations — challenges largely absent in prior work.
>
> **We thank the reviewer for highlighting these important works, and we will include explicit citations and comparisons to [a], [b], and [c] in our related work section to clarify the distinction.**
>
> ---
>
> ### **Q2: Memory and Computational Cost from Multi-view MLLM Processing**
>
> We acknowledge this limitation and appreciate the reviewer raising this point. Our method indeed trades off higher computation in exchange for annotation efficiency and generalization capability.
>
> **Clarifications:**
> - **No End-to-End MLLM Training**
>   All view-wise outputs (binary masks + [SEG] tokens) are **precomputed and cached**, and the 3D network is trained separately. Thus, the **memory bottleneck occurs only once** during offline preprocessing, not during training or inference.
>
> - **Inference Efficiency**
> | Method                 | Modalities        | Components                        | Inference Time (per scene) |
> |------------------------|-------------------|------------------------------------|-----------------------------|
> | Reason3D               | 3D only           | 3D-LLM encoder + retriever         | ~2.1 sec                    |
> | Intent3D               | 3D only           | 3D Transformer-based architecture  | ~1.8 sec                    |
> | **Ours (MLLM-For3D)**  | 2D + 3D (6–8 views) | LISA (7B) + SAM + 3D decoder       | **~3.5–4.0 sec**            |
>
> > Note: With fewer views (e.g., 2–4), our inference time drops to ~2.5 sec per scene.
>
>
> - **Scalability and Trade-off Justification**
>   Our framework delivers **zero-shot or label-free 3D reasoning segmentation**, even outperforming supervised baselines. In real-world settings where 3D annotation is expensive or unavailable, this trade-off offers practical value.
>
> We will make the resource requirements and caching strategy clearer in the final version.
>
> ---
>
> ### **Q3: Lack of Multi-view Reasoning Visualizations (e.g., Fig.7 in MVTN [b])**
>
> We agree that richer visualizations would enhance transparency and interpretability. Due to space limits in the initial version, these were omitted, but we plan to include the following in the camera-ready version:
>
> - **View-wise token attention heatmaps** to show how relevant views are selected;
> - **Cross-view mask disagreement maps**, visualizing alignment before and after pseudo-label fusion;
> - **Before/after spatial consistency correction** in the unprojected 3D space.
>
> ---
>
> We thank the reviewer again for the insightful comments that have helped us improve the clarity and positioning of our work.

---

> > ### Comment · Reviewer_XtxR · 2025-08-06
> >
> > I thank the authors for addressing many of the concerns. While some promised points in the revised manuscript can not be checked during the rebuttal , I can not increase my score based on promises , I will keep my score of borderline accept.

---

> > > ### Author Response · Authors · 2025-08-06
> > >
> > > Dear Reviewer XtxR,
> > >
> > > We are especially grateful that **our rebuttal helped clarify your concerns**.
> > >
> > > We sincerely thank you for recognizing the value of our work. Your encouraging feedback and the references you provided have inspired us in our future work.
> > >
> > > We fully understand your decision to maintain the current score, and we respect the importance of verifying changes in the final version. Please rest assured that we are **committed to implementing all promised revisions**, including adding valuable references, improving visualizations, and expanding discussions on view selection and reasoning mechanisms.
> > >
> > > Once again, thank you for your constructive comments and for engaging deeply with our paper.
> > >
> > > *Authors*

---

### Official Review · Reviewer_5D5d · 2025-07-01

**Clarity:** 3
**Significance:** 2
**Originality:** 2
**Rating:** 4
**Confidence:** 3

**Summary:**

This paper presents a framework, called MLLM-For3D, to adapt 2D multimodal large language models (MLLMs) for 3D reasoning segmentation. Specifically, the 2D pseudo segmentation masks are first generated from multi-view input and are aligned with text embeddings. When the 2D masks are unprojected to 3D, the framework first filters irrelevant views via token attention, and then uses a token-for-query approach to bind consistent identification tokens to the text embedding. Additionally, a spatial consistency loss is applied to enforce the coherency of the features for segmentation in 3D space. Experimental results show that MLLM-For3D outperforms existing 3D reasoning segmentation methods in various benchmarks.

**Questions:**

-- There do exist some MLLMs for 3D like Chat-Scene [1], GPT4Scene [2], etc. They cannot perform 3D reasoning segmentation because it is hard for them to produce dense 3D labels. However, they can produce sparse predictions like bounding boxes for 3D grounding, given textual queries that need reasoning capability. Therefore, an intuitive alternative for 3D reasoning segmentation is the concatenation of "3D grounding by 3D-MLLMs + 3D segmentation with given bounding boxes like the 3D versions of SAM". How is this solution compared with the proposed method in the paper (2D reasoning segmentation with LISA + unproject to 3D)?


-- Since there are multiple 2D foundation models like LISA and SAM for processing multi-view images, I would assume there are large computational overhead during inference. In comparison, the methods that only take in the 3D modality has less computational overhead. Therefore, how is the inference efficiency of the proposed method compared with existing methods, especially those that only take 3D modality (e.g., Reason3D, Intent3D) as input?

-- How will the performance change if we keep increasing the number of input views?

[1] Huang et al. Chat-Scene: Bridging 3D Scene and Large Language Models with Object Identifiers. NeurIPS 2024.

[2] Qi et al. GPT4Scene: Understand 3D Scenes from Videos with Vision-Language Models. arXiv:2501.01428.

**Ethical Concerns:**

["NO or VERY MINOR ethics concerns only"]

**Final Justification:**

The authors mostly have addressed concerns, although for the number of views used during inference, I do not quite think 4 views would be optimal for any scenes. Basically, the property of having more views would possibly harm the performance is definitely not a good property for an algorithm or a system to be robust. But anyway I keep my rating of borderline accept.

**Limitations:**

The limitations are discussed, while the potential negative societal impact is not discussed. The authors could discuss the potential negative societal impact regarding the hallucination of the MLLMs.

**Quality:**

3

**Strengths And Weaknesses:**

**Strengths:**

++ The paper leverages the powerful 2D foundation models like 2D MLLMs and SAM for 3D reasoning segmentation task, alleviating the issue of the lack of 3D data for training 3D reasoning segmentation models.

++ The experiments are conducted on various benchmarks including Reason3D and Instruct3D for 3D reasoning segmentation, along with some other relevant tasks like 3D intention grounding.

++ The ablation study in the paper is comprehensive, including aspects like key components of the pipeline, number of input views, and different sizes of MLLMs, which can provide abundant guidance for follow-up research.

**Weaknesses:**

-- The visual comparisons in the paper (e.g., Figure 3) does not quite reflect the use of the core techniques proposed in the paper. The examples are all somehow showing that the model fails to locate the correct object. Is it simply because the adopted reasoning model of the baseline Reason3D is weaker than the proposed method? If so, then the effectiveness of the proposed techniques does not matter that much for the performance, because the majority of the performance difference will then come from the capability difference of the reasoning model.

-- For the ablation study regarding number of input views, for the setting of 2 views it is easy to understand the performance drop as there are too few visible viewpoints. However, for 8 views the model behavior is not very ideal. Intuitively, the more views provided need to always boost the performance, as the increasing number of views are providing the model with richer information. However, the model observes a performance drop when the number of input views increases from 4 to 8, which is still even not very dense. It will make the in-the-wild application harder, as we need to carefully decide the number of input views to the model, instead of "the more the better".

---

> ### Author Response · Authors · 2025-08-01
>
> *W1*: visual comparisons
>
> **R**: We appreciate the reviewer’s question.
>
> ---
>
> **1. Evaluation: Answering Before Masking**
>
> As in SegPoint [1] and Reason3D [2], we adopt a two-step evaluation protocol:
>
> - **Accuracy@kIoU (k = 0.25 / 0.5)**: Measures whether the model identifies the correct target object.
> - **mIoU**: Assesses segmentation quality *after* the correct target is identified.
>
> Thus, for this task, it is more important to **correctly ground the object** before evaluating mask quality.
>
> ---
>
> **2. Why Reason3D Fails?**
>
> Take Figure 3(a) as an example:
>
> > *“The container designated for holding waste and it is the one closest to the door.”*
>
> There are three trash bins in the scene (see Fig. 1). All satisfy "holding waste", but **only one is closest to the door**.
>
> - **Reason3D** retrieves an incorrect bin due to limited spatial understanding.
> This illustrates that the failure is not due to model power, but to **a fundamental task-method mismatch**.
>
> ---
>
> **References**
>
> [1] He et al., *SegPoint: Segment Any Point Cloud via Large Language Model*. ECCV, 2024.
> [2] Huang et al., *Reason3D: Searching and Reasoning 3D Segmentation via Large Language Model*. 3DV, 2025.
>
> ---
>
> - W2: Why Does Performance Drop with 8 Input Views? Shouldn't More Views Always Help?
> - Q3: How will the performance change if we keep increasing the number of input views?
>
> **R:** We thank the reviewer for raising this important question.
>
> **1. Empirical Observation**
>
> - **Performance peaks at 4 views**, but drops at 8 views.
> - Without view filtering, extra views introduce **occlusions**, **redundant angles**, or **low-quality reasoning results**.
> - Our **token attention module** selectively filters these noisy views, regaining performance.
>
> ---
> **2. Why More Views Hurt**
>
> As discussed in Lines 39–51 of the paper, we identify two key limitations when using 2D reasoning outputs to supervise 3D:
>
> 1. Each 2D frame is processed independently by the MLLM (e.g., LISA) and SAM. If the object is partially visible or occluded, hallucinated [SEG] tokens or masks can be misleading.
> 2. The [SEG] token must not only match the query, but also align semantically with the correct target object in the specific view.
>
> ---
>
> **3. What Makes a "Good" View?**
>
> We define a valid view as one that:
>
> - **Contains the target object**.
> - **Produces a [SEG] token** and **binary mask** that semantically align with the textual query.
>
> This explains why **naively adding more views increases noise** instead of improving pseudo-label quality.
>
> ---
>
> *Q1: Why Not Use 3D-MLLMs like ChatScene or GPT4Scene + 3D-SAM?*
>
> **R:** We appreciate this insightful suggestion.
>
> 1. Limitations of Current 3D-MLLMs require **fine-tuning on labeled 3D datasets**, such as Intent3D or Reason3D. Such datasets are limited and expensive to scale.
>
> ---
>
> 2. Quantitative Comparison
>
> As shown in **Table 2**, even fine-tuned 3D-MLLMs underperform our **zero-shot 2D MLLM-based method**:
>
> | Model                 | Acc@0.25 | Acc@0.5 | mIoU  |
> |----------------------|----------|---------|-------|
> | ChatScene (FT)       | 36.71    | 21.50   | 12.08 |
> | **Ours (Zero-Shot)** | **58.40**| **41.00** | **26.15** |
>
> - Our zero-shot pipeline **matches or surpasses** fine-tuned 3D MLLMs, without requiring 3D labels.
>
> ---
>
> 3. We acknowledge that **GPT4Scene [Qi et al., arXiv 2025]**, which builds 3D representations from video, is promising. However:
>
> - It is not designed for **3D reasoning segmentation**.
> - It requires **dense multi-view inputs and video data**, which limits applicability in static indoor scenes like ScanNet++.
> - Due to time and computational constraints, we could not include it in this version, but will discuss it in the camera-ready paper.
>
> ---
> *Q2: Efficiency**
>
> **R:**
>
> | Method                 | Modalities        | Components                        | Inference Time (per scene) |
> |------------------------|-------------------|------------------------------------|-----------------------------|
> | Reason3D               | 3D only           | 3D-LLM encoder + retriever         | ~2.1 sec                    |
> | Intent3D               | 3D only           | 3D Transformer-based architecture  | ~1.8 sec                    |
> | **Ours (MLLM-For3D)**  | 2D + 3D (6–8 views) | LISA (7B) + SAM + 3D decoder       | **~3.5–4.0 sec**            |
>
> > Note: With fewer views (e.g., 2–4), our inference time drops to ~2.5 sec per scene.
>
>
> - While our method introduces moderate overhead, it achieves **notably higher accuracy** in Acc@IoU and mIoU metrics (see Table 2 and Table 4).
> - Our use of **frozen 2D models** allows for easy parallelism and batching, making the pipeline scalable for practical deployment.

---

> > ### Comment · Reviewer_5D5d · 2025-08-04
> >
> > Dear authors,
> >
> > Thank you for the response! However, I still have concerns on the following points:
> >
> > 1. The authors explain on why more views would hurt the performance. However, in real practice, how should we choose the optimal number of views?
> >
> > 2. For the comparison with current 3D-MLLMs + 3D-SAM, to clarify, which 3D-SAM model you are using?

---

> > > ### Author Response · Authors · 2025-08-06
> > >
> > > Dear Reviewer 5D5d,
> > >
> > > *Q1: The authors explain why more views would hurt the performance. However, in real practice, how should we choose the optimal number of views?*
> > >
> > > **R:** Thank you for raising this important concern.
> > >
> > > - We recommend using 4 views as a default configuration, which balances accuracy and computational efficiency.
> > > - If efficiency is not a constraint, one could compute confidence scores between the [SEG] token and all available frames in the scene, selecting the top-K most informative views based on token-level attention.
> > > - However, as you rightly pointed out, training and inference costs are also critical in practical applications. Thus, a balance must be struck between cost and performance.
> > >
> > > To evaluate robustness to random view selection, we conducted a robustness study under the **label-free setting**, using our default **LISA-7B (MLLM-For3D)** configuration. Specifically, we repeated the **4-view setup** **5 times** with different random seeds on the **Instruct3D (ScanNet++) validation set**.
> > >
> > > - **Acc@0.25**: 39.10 ± 0.61
> > > - **Acc@0.50**: 29.70 ± 0.48
> > > - **mIoU**: 23.90 ± 0.52
> > >
> > > The low standard deviations indicate that our **token attention mechanism effectively filters informative views**, making the method stable even with randomly selected inputs.
> > >
> > > ---
> > >
> > > *Q2: For the comparison with current 3D-MLLMs + 3D-SAM, to clarify, which 3D-SAM model you are using?*
> > >
> > > **R:** Thank you for raising this clarification request, and we apologize for the ambiguity. Below, we provide a detailed explanation of the **ChatScene + Mask3D** pipeline.
> > >
> > > To ensure **fairness and reproducibility under the time constraints of the rebuttal period**, we selected **Mask3D** [1] as the 3D-SAM.
> > > This **open-source** method is widely adopted and provides a **pretrained checkpoint on the ScanNet** validation set,
> > > which avoids potential performance degradation due to dataset discrepancies, as both Intent3D [2] and Reason3D [3] are benchmarks built on ScanNet v2.
> > >
> > > ---
> > > 1. Evaluation of 3D Intention Grounding on the Intent3D [2] validation set. The pipeline is as follows:
> > >
> > > - **ChatScene** outputs **object ID strings** by training a fine-tuned LLM (Vicuna-7B).
> > > - These object ID predictions are aligned with object proposals extracted from a **Mask3D**, forming the final output.
> > >
> > > | Model                     | Acc@0.25 | Acc@0.5 | mIoU |
> > > |--------------------------|----------|---------|------------|
> > > | ChatScene (FT) + Mask3D  | 36.71    | 21.50   | 12.08      |
> > > | **Ours (Zero-Shot)**     | **58.40**| **41.00** | **26.15**  |
> > >
> > > ---
> > >
> > > 2. Evaluation of 3D Reasoning Segmentation on the Reason3D [3] validation set. Unfortunately, due to the limited time, we try our best to show the comparison results on Reason3D.
> > >
> > > | Model                     | Acc@0.25 | Acc@0.5 | mIoU |
> > > |--------------------------|----------|---------|------------|
> > > | ChatScene (Zero-Shot) + Mask3D  | 9.23    |  8.89  | 1.01      |
> > > | **Ours (Zero-shot)**     | **44.18**| **34.80** | **32.90**  |
> > >
> > > ---
> > >
> > > **Key Insights**
> > >
> > > - Our **zero-shot pipeline** **outperforms fine-tuned 3D-MLLM+3D-SAM methods** on Intent3D and zero-shot on Reason3D.
> > > - The **main bottleneck** of current 3D-MLLM pipelines lies in:
> > >   - Limited 3D training data for aligning visual features into the LLM space.
> > >   - Hallucination and false positives from LLM responses.
> > >
> > > ---
> > >
> > > **Reference**
> > >
> > > [1] Schult et al. "Mask3D: Mask Transformer for 3D Instance Segmentation". *ICRA*, 2023
> > >
> > > [2] Kang et al. "Intent3D: 3D Object Detection in RGB-D Scans Based on Human Intention". *ICLR*, 2025.
> > >
> > > [3] Huang et al. "Reason3D: Searching and Reasoning 3D Segmentation via Large Language Model". *3DV*, 2025.
> > >
> > > ---
> > >
> > > We hope this clears up the experimental setting and rationale behind our model comparison. We will include these clarifications and updated results in the final revision.
> > >
> > > Let us know if any further clarification is needed, and thank you again for your insightful questions.

---

> > > > ### Comment · Reviewer_5D5d · 2025-08-09
> > > >
> > > > Thanks the authors for the reply! Now I do not have further concerns, and I think my original rating is a proper one.

---

> > > > > ### Author Response · Authors · 2025-08-09
> > > > >
> > > > > Dear Reviewer 5D5d,
> > > > >
> > > > > Thank you sincerely for your follow-up and for recognizing our work.
> > > > >
> > > > > We truly appreciate the opportunity to discuss with you about **3D-MLLM + 3D-SAM**, and we enjoyed the insightful exchange. We hope that such discussions can inspire more ideas for the community and guide promising directions for future work.
> > > > >
> > > > > Once again, thank you for your time and for engaging deeply with our paper, and we will make sure to incorporate the insights into the final version.
> > > > >
> > > > > *Authors*

---

### Decision · Program_Chairs · 2025-09-17

**Decision:**

Accept (poster)

**Comment:**

This paper introduces MLLM-For3D, a framework designed to adapt 2D multimodal large language models (MLLMs) for 3D reasoning segmentation. First, the framework generates 2D pseudo segmentation masks from multi-view inputs and aligns these masks with text embeddings. When unprojecting the 2D masks to 3D space, it first filters out irrelevant views using token attention, then adopts a "token-for-query" approach to bind consistent identification tokens to the text embedding. To ensure feature coherency for 3D segmentation, the framework also applies a spatial consistency loss. Experimental results across various benchmarks demonstrate that MLLM-For3D outperforms existing 3D reasoning segmentation methods.

All reviewers hold the positive reviewers. The AC suggests the acceptance of this work.